# Effects of Extraction Temperature of Protein from Date Palm Pollen on the Astringency Taste of Tea

**DOI:** 10.3390/foods14030508

**Published:** 2025-02-05

**Authors:** Rania Mohamed, Jizhou Xie, Fang Wei, Liyong Luo, Wei Luo, Liang Zeng

**Affiliations:** 1Integrative Science Center of Germplasm Creation in Western China (CHONGQING) Science City, College of Food Science, Southwest University, Chongqing 400715, China; raniamohammed14@yahoo.com (R.M.); xiejizhou741@126.com (J.X.); wf2937093615@126.com (F.W.); lly1979@swu.edu.cn (L.L.); 2Department of Food Science and Technology, Faculty of Agriculture, University of Khartoum, Shambat 13314, Sudan; 3Chongqing Tea Technology and Innovation Center, Chongqing 400715, China; 4Chongqing Key Laboratory of Speciality Food Co-Built by Sichuan and Chongqing, Chongqing 400715, China

**Keywords:** protein structure, (-)-epigallocatechin gallate, interaction, spectrometry measurements, sensory evaluation

## Abstract

The astringency of tea, predominantly attributed to epigallocatechin gallate (EGCG), plays a crucial role in shaping its overall quality, and plant-based proteins are gaining popularity as a preferred alternative to milk-based proteins for enhancing the flavor profile of tea. This study investigated the impact of extraction temperature on date palm pollen (DPP) protein quality and tea astringency, comparing temperatures of 30 °C and 80 °C. Results indicated that higher extraction temperatures yield more protein and improve the thermal and surface properties of DPP. The molecular interaction between DPP and EGCG was investigated in an aqueous solution, and spectroscopic analyses (FTIR, UV, and CD) revealed that EGCG interactions at a 1:1 molar ratio induced structural changes in α-helix and β-sheet content in secondary structures in DPP, particularly at 80 °C, which strengthened and enhanced the hydrophobic interactions and hydrogen bonds between DPP molecules as EGCG concentration increased. A sensory evaluation using quantitative descriptive analysis (QDA) confirmed a significant reduction in astringency in DPP–tea polyphenol solutions extracted at 80 °C. This research highlights the potential of DPP as a functional ingredient in the food industry, creating a protein-polyphenol complex that reduces tea’s astringency while maintaining its unique flavor profile, thus offering a novel approach to enhance tea beverages.

## 1. Introduction

The health benefits of tea are widely recognized all over the world, but the enduring bitterness and astringency, especially in green tea, remain significant challenges to broader acceptance. These sensory qualities are primarily linked to tea’s polyphenol-rich composition, encompassing flavonoids, phenolic acids, and alkaloids [1]. And catechins, particularly epigallocatechin gallate (EGCG), which is prevalent in tea leaves, significantly impact the taste profile [2].

Proteins are commonly used to decrease the bitter and astringent taste of tea. Numerous studies have examined the relationship between tea polyphenols (TPs) and proteins [3]. According to [4], TPs have been observed to interact with proteins that include a large concentration of alkaline amino acids, which can cause the peptide tertiary structure to loosen and expose hydrophobic groups. Because of their abundance in hydroxyl groups, catechins can create hydrogen bonds with proteins’ nitrogen and oxygen [5,6]. According to most theories, there are three steps to the interaction between TPs like EGCG and proteins [7]. First, a loose, randomly coiled conformation protein binds to several sites on polyphenols, causing the protein to shrink physically and take on a tighter, spherical structure. The polyphenol complex binds to the protein surface in the second step as the concentration of polyphenols increases. Protein molecules dimerize one another, which lowers their solubility. The final stage involves the dimers’ spontaneous assembly, which results in a significant quantity of particle precipitation [8].

Milk is commercially used to decrease the bitter and astringent taste of tea; however, plant-based proteins are increasingly favored over milk-based proteins primarily because of their lower environmental impact, health benefits, and suitability for various dietary needs [9]. Moreover, plant-based proteins are typically easier to digest and are free from animal-derived hormones and antibiotics.

The pollen of the date palm (*Phoenix dactylifera* L.), a member of the Aceraceae family, contains more than 30% of protein [10]. And it has been widely utilized as a medicinal remedy by the ancient high Egyptians and Chinese.

Processing factors like heat treatment can influence the interactions between polyphenols and proteins, as well as the structure of proteins and the biological activities of polyphenols. Many authors have noted that high heat treatments can improve the strength of polyphenol–protein interactions [11]. This raises intriguing questions about the protein part and how different extraction temperatures affect its properties. However, it is known that all proteins denature at temperatures over 40 °C to 60 °C. Depending on the specific proteins present and the experimental conditions, this temperature varies; higher temperatures can ensure inactive enzymes or disrupt protein–protein interactions. Therefore, in this study, we employed two distinct temperatures to guarantee the interaction between protein and polyphenols.

This study is the first to explore how to extract protein from DPP at different temperatures. The goal is to reduce the astringency and bitterness of tea by using the interactions between protein and polyphenols. Two previous investigations examined the efficacy of DPP concentrate [12,13]. These studies looked at how to extract protein from DPP using ultrasound at 25 °C for 2 s, followed by separation using isoelectric precipitation at 30 °C. The results showed that the protein obtained using the ultrasound method had better surface activity compared to the conventional extraction method. This protein could be used as a natural surfactant in the agri-food and pharmaceuticals fields. To our knowledge, no one has previously defined the interaction between DPP and EGCG, particularly in reducing tea astringency, and the mechanism underlying this interaction remains unknown.

The objective of this study was to explore the influence of heat treatment at 30 °C and 80 °C on the quality of DPP protein based on its physio-chemical properties. Furthermore, the aim is to analyze the interactions between tea polyphenols (specifically EGCG) and DPP protein and to assess their potential application in the food industry as a functional beverage with a novel flavor profile. The determination of binding kinetics was carried out using particle size and ultraviolet–visible (UV–Vis) spectroscopy, while circular dichroism and FTIR were employed to measure the impact of protein structure on binding. Additionally, a sensory evaluation (QDA) was utilized to investigate how this interaction influenced the astringent and bitter flavors of the tea. Ultimately, the findings from this study may serve as a valuable foundation for developing plant-based alternatives for popular food products that possess similar physical and chemical attributes to their traditional counterparts.

## 2. Materials and Methods

### 2.1. Materials

The DPP powder was procured from Xi’an Xuanqing Import & Export Co., Ltd. (Xi’an, China) and maintained at 4 °C until required. EGCG was provided by Shanghai Darui Fine Chemistry Co., Ltd. (Shanghai, China). and subsequently stored at 4 °C for future use. All remaining chemicals utilized were of analytical grade, unless specified otherwise. Sodium hydroxide and hydrochloric acid were purchased from Yuexiang Chemical Industry Company Limited (Chongqing, China). Tannin and quinine monohydrochloride dehydrate were obtained from Bide Company (Shanghai, China).

### 2.2. Proximate Composition and Bio-Active Compounds of Date Palm Pollen

#### 2.2.1. Chemical Composition Analysis

This study aimed to determine moisture, crude fiber, crude fat, crude protein, and ash in a sample using the official AOAC method [14]. Moisture was determined by weighing two grams of well-mixed samples in a preheated dish and heating them in an oven for 24 h. Ash was determined by placing the sample in a crucible and heating it in a muffle furnace for 3 h or more until white gray or reddish ash was obtained. Crude protein was determined using the micro Kjeldahl (Qingdao Shenghan Chromatograph Technology Co., Ltd., Qingdao, China) (N × 6.25) distillation method, which involved digesting the sample with a catalyst mixture and adding concentrated nitrogen-free sulfuric acid. Fat (ether extract) was determined via a Soxhlet extractor using a dry, empty extraction flask, and petroleum spirit. Crude fiber was determined using a mixture of H_2_SO_4_ and pre-heated KOH, and carbohydrates were determined by difference using the equation: 100 − [moisture + crude fat + crude protein + ash + crude fiber] %.

#### 2.2.2. Polyphenols Extraction and Determination

A total of 2 g of DPP were combined with 20 mL of 50% acetone and vigorously stirred for 2 h at 25 °C before being centrifuged for 20 min at 4500 rpm. To enhance the extraction of polyphenols and flavonoids, the process was repeated twice [15].

The Folin–Ciocâlteu method was utilized to determine the total phenolic content. A combination of 500 µL of the previously mentioned extract and 2.5 mL of a 1:10 Folin–Ciocâlteu solution in water was prepared, followed by the addition of a 2 mL solution of 7.5% sodium carbonate. Subsequently, the tubes were left at room temperature for 15 min, and the absorbance was measured at 765 nm using a UV spectrophotometer (Shimadzu, Tokyo, Japan), with distilled water used as the blank sample [16]. The total polyphenol content was quantified in mg/100 g DPP and expressed as gallic acid equivalents (GAEs). A standard curve using gallic acid with concentrations ranging from (0 to 50) mg/L was established [15].

#### 2.2.3. Determination of Flavonoids

The process involved transferring the previously mentioned extract (250 µL) into a tube with 1 mL of distilled water and 150 µL of NaNO_2_ 15%. After 6 min, 75 µL of AlCl_3_ 10% were introduced. Subsequently, 1 mL of 1 mol/L NaOH was added, and the final volume was adjusted to 2.5 mL with distilled water. Following a 15 min incubation, the absorbance was gauged at 510 nm against distilled water as the blank sample using a UV spectrophotometer (Shimadzu, Tokyo, Japan). The content of flavonoids was denoted as quercetin equivalents (EQ) in mg/100 g DPP [17].

### 2.3. Preparation of Date Palm Pollen Protein Concentrates by Isoelectric Participate Method

A mixture of 1:10 (*w*/*v*) DPP combined with distilled water and adjusted to a pH of 12 using 1 mol/L NaOH was subjected to magnetic stirring using an HWCL-1 thermostatic magnetic stirrer (Zhengzhou Greatwall Scientific Industrial and Trade Co., Ltd., Zhengzhou, China) for 2 h at temperatures of 30 °C and 80 °C. Afterward, a 30 min centrifugation at 10,000× *g* at 4 °C was carried out to obtain pollen protein concentrate through the isoelectric precipitation method. To ensure maximum protein recovery, the pellet underwent double extraction, and the resulting supernatants were collected for subsequent precipitation. Then, the pH of the supernatant was lowered to 3 with 1 mol/L HCl and incubated at 4 °C overnight. The protein concentration was achieved by centrifugation at 10,000× *g* for 30 min at 4 °C. The resulting pollen protein concentrate was neutralized, dialyzed against ultra-pure water for 7 days (molecular weight cutoff: 10 KDa), and freeze-dried (Ningbo Xinzhi Scientz-30ND freeze-drying machine- Ningbo Scientz Biotechnology Co., Ltd., Ningbo, China) for 48 h. The samples were then stored at −80 °C until use [12].

### 2.4. Determination of Date Palm Pollen Protein Properties

#### Determination of Extraction and Protein Yields

The quantification of extraction yield for each sample involves a mathematical representation of the amount of product acquired from the initial amount of DPP. The determination of protein yield was derived from the protein content and the weight of the extract, taking into account the protein content and the initial weight of DPP.

### 2.5. Scanning Electronic Microscopy (SEM)

Scanning electron microscopy (SEM) is widely considered an unparalleled imaging technique of interest. This device operates by scanning a beam of electrons onto a specimen, causing it to view fine surface details of the specimens. The DPP protein sample preparation for SEM was spread on a conductive adhesive carbon strip that was fastened to the stub and covered with a thin coating of gold using (fine coat, JFC-1100 E, Ion sputtering device, JEOL, Tokyo, Japan). In order to effectively produce images through SEM (JSM-5400, JEOL, Tokyo, Japan), the microstructure was examined at an accelerating voltage of 15 kV and a magnification of 500 and 2000.

### 2.6. Differential Scanning Calorimetry (DSC)

The application of differential scanning calorimetry (DSC) involves the utilization of an isothermal analysis instrument (TA DSC Q1000, New Castle, DE, USA). A quantity of 2 mg of DPP protein samples was placed into sealed aluminum pans, while an empty pan of equal mass was used as a reference. Subsequently, the heat flow rate was controlled within the range of 50 °C to 250 °C at a scan rate of 5 °C/min under a nitrogen environment. This heat flow rate is equivalent to the rate of temperature change, any heat flow resistance present within the sample, and the rate at which the sample generates or absorbs heat. The experiment was conducted in triplicate.

### 2.7. Surface Tension Measurement

The capillary rise method serves as a renowned technique for surface tension measurement. In the dynamic mode, the surface tension was measured using the automated drop volume tensiometer (TVT1, Lauda, Germany). To ascertain the surface tension of DPP, two concentrations (0.5% and 1% of protein/100 mL) of each sample were examined. The solutions were prepared at pH 7 (adjusted by 1 mol/L NaOH) the day before the experiment. The syringe volume was almost 3 mL. All measurements were conducted in triplicate at room temperature.

### 2.8. Sensory Evaluation

The QDA technique was utilized to examine the impact of DPP (30 and 80 °C) inclusion on the astringency and bitterness of the TP solution. Trained evaluators, consisting of 2 males and 8 females aged 20–30 years, from the School of Food Science at Southwest University (Chongqing, China), performed the sensory assessment. Prior to participating, each individual provided written informed consent. For the evaluation samples, the TP concentration remained constant at 0.18 g/L, while the DPP concentrations at 30 °C and 80 °C were 0.18 g/L, 0.27 g/L, 0.36 g/L, and 0.45 g/L. The samples were dissolved at room temperature and filtered through Whatman No. 1 filter paper. Each evaluator received nine sequentially labeled cups of sample solution, each approximately 10 mL. Before tasting each sample, the evaluators rinsed their mouths with pure water, then swirled the sample in their mouth for (8–10) s, rating the astringency and bitterness before spitting out the solution. After scoring each sample, the evaluators were instructed to rinse their mouths with pure water. A 6 min interval was observed between each sample test to minimize sensory fatigue and carry-over effects. Evaluators were asked to rate the astringency and bitterness taste intensities of each solution using a five-point scale, ranging from “extremely strong” to “extremely weak”. Tannin served as the standard reference for astringency, while quinine monohydrochloride dehydrate was the standard reference for bitterness. Each standard solution was presented in various concentrations corresponding to different points on the scale. Two (1 h) sessions were conducted, with nine samples tested in each session, and each sample was repeated twice.

### 2.9. Preparation of Protein Solution and Tea Polyphenol Solution

#### Particle Size Measurements

Different amounts of EGCG were amalgamated with DPP protein to formulate combinations with EGCG concentrations of 100 μM, 200 μM, 300 μM, and 400 μM while sustaining the DPP protein content at 10 μM. Using the technique described by Yang et al. [18], the mixture was kept at 25 °C for 2 h, and the development of haze was tracked using dynamic light scattering (DLS). Utilizing DLS techniques is a prevalent method for determining the size of particles in a given sample. We used the Zetasizer Nano-ZS90 Malvern machine, Worcestershire, UK, to measure the size and spread of EGCG/DPP complexes by checking the amount of light that was scattered at a 90-degree angle using dynamic light scattering (DLS). The results were represented in terms of the polydispersity index (PdI) for size distribution and the cumulative mean diameter (nm) for particle size. To ensure precision, this experiment was repeated three times.

### 2.10. Fourier Transform Infrared (FTIR) Spectroscopy

FTIR spectroscopy instruments quantitatively evaluated samples using infrared light, following the method of Yang et al. [18] with minor modifications. Prior to the scan, EGCG/DPP complexes were prepared in a 1:1 molar ratio of DPP protein to EGCG. Subsequently, they were freeze-dried and combined with potassium bromide (KBr). The FTIR spectroscopy of DPP protein and the EGCG/DPP complex was conducted using a Spectrum 100 FTIR instrument (Perkin-Elmer, Shelton, UK) within the 400–4000 cm^−1^ range, at a resolution of 4 cm^−1^. Twenty scans were conducted for each measurement. The analysis was conducted utilizing Origin 8.0 software.

### 2.11. Circular Dichroism (CD) Measurement

Circular dichroism (CD) spectra were obtained using a spectropolarimeter (MOS-500, Cecine Parisse, Paris, France) for a DPP protein sample at a concentration of 0.5 g/L, following its mixing with EGCG. Measurements were conducted with a light source spanning 190–400 nm wavelength at 25 °C, utilizing a 1 mm path-length cell. The scanning speed was established at 100 nm/min, with a bandwidth of 1.0 nm. The analysis of the sample was conducted using Origin 8.0 software.

### 2.12. UV–Vis Absorption Spectra

The EGCG/DPP aqueous extract was prepared at a fixed concentration of 4.0 × 10^−5^ mol/L for DPP protein, while EGCG was introduced in increasing concentrations of 0 × 10^−5^ mol/L, 2.0 × 10^−5^ mol/L, 4.0 × 10^−5^ mol/L, 6.0 × 10^−5^ mol/L, 8.0 × 10^−5^ mol/L, and 10.0 × 10^−5^ mol/L. Measurements were conducted using a UV–visible spectrophotometer (Model UV-2450 Shimadzu, Tokyo, Japan) over a wavelength range of 190–400 nm at 25 °C, utilizing a 1 cm quartz cell [19].

### 2.13. Statistical Analysis

The analytical values were obtained from three separate measurements. The results were presented as mean values ± standard error from three separate measurements. The statistical analysis, data processing, and mapping were performed utilizing the software programs Origin software 22 and Minitab software 22 to compute variance (ANOVA) and conduct Tukey’s multiple comparisons.

## 3. Results and Discussion

### 3.1. Chemical Composition and Bio-Active Components

#### 3.1.1. Chemical Composition

This study involved an analysis of moisture, fiber, ash, crude fat, total protein, and carbohydrate content for palm pollen powder. Table 1 presents the detailed chemical composition of DPP powder. Our analysis indicated that protein, carbohydrate, and fat constituted 34.42%, 32.88%, and 20% of the total, respectively [10,20,21]. These researchers presented analogous findings: the protein content ranged between 31.11, 32.95, and 30.87%; the carbohydrate content ranged from 13.41 to 35.06%; and the fat content ranged from 19.31 to 20.74%. Compared to date fruit (1.1 to 3.0 g/100 g) and seeds (2.29 to 7.08 g/100 g), DPP demonstrated a superior protein concentration (15.8 to 38.18 g/100 g) [22]. Hassan [10] identified palmitic (C16:0), linoleic (C18:2), and myristic (C14:0) acids as the predominant fatty acids in DPP grains. Our data unequivocally demonstrate that DPP functions as a significant source of proteins employed in the agri-food and medicinal industries.

#### 3.1.2. Bio-Active Components of DPP

According to the findings in Table 1, the phenol content of DPP was ascertained to be 227.46 mg GAE/100 g, which is inferior to the polyphenol content of DPP documented by [15] (909.30 mg GAE/100 g) and [23] (213.36 to 197.62 mg GAE/g). This finding confirms that DPP is a good source of bioactive chemicals that are found naturally and can effectively stop oxidation in a wide range of food products. Also, the amount of flavonoids in DPP was measured to be 6.37 mg EQ/100 g. This is similar to the value found by [15] (4.31 mg EQ/100 g) but much lower than the 266 mg EQ/100 g found in DPP by [24] and the 168 mg EQ/100 g found by [25]. The overall concentration of flavonoids and phenolic chemicals may vary depending on a variety of factors, including storage, drying methods, and geographical origin. Also, the choice of extraction solvents is likely to have a big effect on how well the phenolic component is recovered [15].

### 3.2. Effect of Temperature on Protein Extraction and Yield from DPP

Figure 1 shows that the protein yield varied significantly (*p* < 0.05) by heat treatment during the isoelectric precipitation procedure of DPP at 30 °C and 80 °C. Temperature at 80 °C significantly improved protein yields and extraction. As can be shown, for DPP 30°C and DPP 80 °C, respectively, the protein yield increased from 25.58% to 29.07%, and the extraction yield increased slightly from 11.49% to 12.50%. This occurred because rising temperatures cause disruptions in the cell wall, which facilitate the release of various compounds; refs. [12,13] reported a similar finding. The weight obtained for each concentrate was mainly responsible for this difference.

### 3.3. Scanning Electron Microscopy (SEM)

SEM analysis was employed to investigate the microstructure of the freeze-dried extracts to elucidate the effect of temperature on DPP protein. Figure 2 illustrates the variations in the protein structure of DPP at 30 °C and 80 °C. At magnifications of 500× and 2000×, the aggregate sizes (about 100 and 30 μm) exhibited morphological variations in the protein bodies of the temperature isolates.

The detected entities exhibited various sizes, occasionally round and at other times oval, characterized by a smooth surface and many indentations. The comparison of DPP at 30 °C (Figure 2(A2)) with DPP at 80 °C indicated that DPP at 30 °C exhibited several smaller entities encircling the bigger ones, but DPP at 80 °C had predominantly larger entities, as illustrated in Figure 2(B2). Samples treated at 30 °C demonstrated enhanced homogeneity, resulting in the formation of a protein framework. Temperature influences the dimensions of the final product’s entities; higher temperatures lead to augmented charges and the development of larger clusters during freeze-drying [13]. Treatment at 80 °C produces more irregular fragments and disordered structures. These findings corroborate the concept that thermal treatment can modify protein structure [12].

### 3.4. The Effect of Temperature on Differential Scanning Calorimetry (DSC)

A differential scanning calorimeter is a useful tool for checking how stable proteins are at high temperatures. It makes it simple to quickly and accurately check the heat resistance of any extract. It provides critical information regarding the denaturation enthalpy (ΔH) and denaturation temperature (Td) of the samples. Td is a measure of thermal stability, while ΔH is the quantity of energy needed to denaturize a protein. Figure 3A,B demonstrated that both extracts (30 °C and 80 °C) displayed distinct endothermic peaks. The addition of heat caused the Td for DPP 30 °C and DPP 80 °C to rise from 176.80 °C to 202.59 °C, respectively, and the H to drop from 178.86 J/g to 128.15 J/g. These findings indicate the considerable heat resilience of both extracts. Karra et al. [13] observed similar findings for the protein concentrations in DPP. The observed discrepancies (*p* < 0.05) between the two samples can be attributed to the differing compositions of the concentrates. A protein with very little water changes its Td value and enthalpy because a higher ΔH value means a more ordered structure [26]. When the water content drops to less than 20%, heating stabilizes the protein. According to the discussion in the section on scanning electron microscopy, we can conclude that in our case, heating probably caused a modification in the structure of the pollen protein. More research has shown that the treated sample had broken intermolecular bonds, which led to the release of its hidden negative charges and a drop in ΔH [12].

### 3.5. The Effect of Temperature on Surface Tension

We evaluated the surfactant efficacy of an extract by measuring its surface tension. Figure 4 depicts the behavior of DPP solutions at 30 °C and 80 °C with protein concentrations of 0.5 g/100 mL and 1 g/100 mL, respectively. We observed variations irrespective of the concentration. DPP at 80 °C demonstrated considerably (*p* < 0.05) enhanced surface activity at a concentration of 1 g/100 mL in comparison to DPP at 30 °C. Also, the surface tension of pollen protein concentrations changed as the protein content in dispersions went from 0.5 g/100 mL to 1 g/100 mL while keeping the pH the same. The measurements of the equilibrium surface tension ranged from 44.49 mN/m to 47.69 mN/m. These results were similar to those found for biosurfactants by [13]. DPP at 30 °C exhibited comparable values for 0.5 g/100 mL and 1 g/100 mL protein dispersions, in contrast to DPP at 80 °C. Several parameters, including concentrations and pH, influence the dispersion’s ability to reduce surface tension, as all surface tension measurements demonstrate. DPP at 80 °C exhibited enhanced surfactant characteristics, perhaps because of the participation of other agents beyond proteins. Therefore, pollen protein concentrates could serve as natural surfactants in the agricultural and food sectors, potentially enhancing the sensory qualities of food systems [12].

### 3.6. Descriptive Sensory Evaluation

The aim of the sensory QDA assessment was to identify the tea with DPP addition that exhibited the lowest astringency and would attract the typical consumer. Figure 5 depicts the sensory assessments of the tea polyphenol with TP and DPP protein additions regarding astringency and bitterness, evaluated by 10 panelists. A standardized solution was employed to evaluate the astringency and bitterness of the samples on a scale of 8 to 10, 6 to 8, 4 to 6, 2 to 4, and 0 to 2, denoting “extremely strong”, “strong”, “neutral”, “weak”, and “extremely weak”, respectively.

The astringency scores were considerably affected (*p* < 0.5) by the incorporation of different quantities of DPP at 30 °C and 80 °C to TP; however, there was no significant alteration in the bitterness values (*p* > 0.05). Figure 5A illustrates the sensory score results for bitterness and astringency in TP/DPP at 30 °C and 80 °C at several concentrations (0.18, 0.27, 0.36, and 0.45) g/L (data reported in Appendix A). The graph demonstrates that at 80 °C concentrations, astringency had a negative correlation, whereas at 30 °C, it showed variability. The radar plot (Figure 5B) indicated that sample TP/DPP (0.18 g/L) at 30 °C had a high similarity to TP regarding astringency. Furthermore, sample TP/DPP (0.45g/L) at 80 °C attained the best acceptance rating for astringency mitigation.

### 3.7. Analysis of the Interaction Between DPP and EGCG

According to Yang et al. [18], light-scattering measurements using DLS could provide useful information about the change in quaternary structure in proteins. Therefore, at temperatures of 30 °C and 80 °C, DLS can evaluate the complexes formed by the direct interaction of EGCG with DPP. The DLS data reveal that the compaction of DPP molecules occurs within the first minute after mixing and then stabilizes. The results show an anticipated initial shrinkage at a low EGCG ratio, then an increase in particle size due to aggregation. We used DLS to measure the average size and polydispersity of the EGCG/DPP complexes at 30 °C and 80 °C (Figure 6) (data reported in Appendix A). We noted that the size of the complexes remained unchanged when the EGCG concentration was less than 0.2 mM. Nevertheless, when the EGCG concentration was above 0.2 mM, it significantly (*p* < 0.05) enlarged the size of the complexes, especially in the instance of DPP at 80 °C. Our study showed that we could only look at soluble EGCG/DPP complexes when the EGCG level dropped below 0.2 mM. This means that the molar ratio of EGCG to DPP was 1:1. The results showed that date palm pollen protein extracted at 80 °C may be more susceptible to EGCG cross-linking, which could cause it to aggregate and grow in particle size. Cheng et al. [27] found that a high concentration of polyphenols in tea could cause myofibrillar proteins (MPs) in pork to aggregate, which is in agreement with this result.

### 3.8. FTIR-ATR Analysis

The FTIR spectra of the freeze-dried DPP protein (30 °C and 80 °C) and the complex to EGCG (1:1) were compared, revealing peaks that originated in the 4000–400 cm^−1^ region. These peaks are shown in Figure 7A,B. The DPP30 °C and DPP80 °C spectra are nearly the same, with a few peaks changing. In each spectrum, a major peak corresponding to the hydroxyl group bands O-H, which range from 3402 cm^−1^ for 30 °C to 3410 cm^−1^ for 80 °C, and the EGCG/DPP complex, which peaks at 3264 cm^−1^, was shifted to 3228 cm^−1^ at 30 °C and 80 °C, respectively, indicating the heating treatment and EGCG addition involved intermolecular hydrogen bonds in DPP protein [28]. The C-H stretching vibrations correspond to the band between 3000 cm^−1^ and 2800 cm^−1^, and the changes in this region showed the hydrophobic interaction between EGCG/DPP protein (30 °C and 80 °C). The C=O stretching vibrations of amide I (α-helical structure) were found to occur in the wide range (80%) between 1660 cm^−1^ and 1540 cm^−1^ for pure DPP protein at 30 °C and 80 °C, but in EGCG/DPP (30 °C and 80 °C), the peak of amide I band decreased slightly and ranged between (1620 and 1520) cm^−1^. Thus, the decreased peak indicated the formation of complex. This band is widely used in protein secondary structural analysis. Because the amide II band (β-sheet) is mostly formed from the N-H (20%) bending vibration that is strongly related to C-N stretching, the amide II band of nearly 1550 cm^−1^ is very helpful. Meanwhile, peak regions between 1350 cm^−1^ and 1200 cm^−1^ displayed amide III’s C-H stretching vibrations and N-H deformation. Carbohydrates mainly occupy the spectral range of 1200 cm^−1^ and 900 cm^−1^ [29]. The fingerprint zones of both extracts 30 °C and 80 °C, ranging from 1500 cm^−1^ to 500 cm^−1^, showed differences in protein structure between DPP protein alone and EGCG/DPP, where both peaks have disappeared at 642 cm^−1^ and 634 cm^−1^ found in DPP at 30 °C and 80 °C, respectively, which confirmed the formation of complexes. This might be because the high levels of certain amino acids in DPP protein or the high temperatures used for extraction break up the bonds between molecules [15]. These amino acids are known to disrupt the regular structures of proteins, effectively forming turns and hydrophobic regions that align with the FTIR results. This means that the highly polymerized polyphenols in tea are more likely to interact with the hydrophobic parts of proteins because they have more conjugated and galloylated structures [30]. Yu et al. [31] documented a comparable observation.

### 3.9. Circular Dichroism Analysis (CD)

Far–UV CD spectroscopy was employed to evaluate the effect of EGCG/DPP complex formation on the secondary structure of DPP at temperatures of 30 °C and 80 °C. Figure 8 illustrates that the distinctive α-helix conformation of the protein generated discernible negative peaks at 208 nm and 207 nm for DPP at 30 °C and 80 °C, respectively, and at 212 nm for EGCG with DPP at the same temperatures. The interaction between EGCG with DPP led to a minor decrease in band intensity, suggesting that EGCG modified the composition of the protein’s helical structure, encompassing the α-helix, β-sheet, and random coil configurations. Prior work indicates that a predominant α-helix conformation fosters a compact protein structure, whereas a significant random coil conformation may result in a more relaxed protein structure [32]. The results of this study indicate that EGCG may influence the secondary structure of DPP, resulting in a more relaxed protein conformation. Nevertheless, another study [33,34] has demonstrated that the inclusion of polyphenols enhances the α-helix content of proteins, but the precise mechanism underlying this enhancement is still to be elucidated. The variability in secondary structural alterations can be ascribed to the distinct interferences arising from the composition and binding mechanism of EGCG in the establishment of hydrogen and disulfide bonds within proteins. This observation is consistent with the findings of [35,36] in their studies on pork and chicken protein.

### 3.10. Ultraviolet–Visible (UV–VIS) Spectrophotometry

Utilized a UV–visible spectrum to study the protein structure and how it interacts with ligands. It is a quick, easy, and useful way to find changes in structure and the stage of complex formation [37]. The EGCG/DPP protein complexes can absorb UV light because of changes in electrons in their molecular orbitals (MOs), mostly in their aromatic rings. Researchers [38] primarily attribute the absorption peaks around 280 nm in the UV–visible spectra to aromatic amino acids. Figure 9 illustrates the DPP absorption spectra at 30 °C and 80 °C, both in the presence and absence of EGCG (data reported in Appendix A). According to the results, there was a blue shift and an increase in the absorption peak as the EGCG concentration rose. The findings suggest that EGCG might make DPP’s peptide chain longer, which would reveal its aromatic hydrophobic groups and change the shape of DPP. Haslam et al. [39] stated that the increased absorption of UV light by EGCG indicated the formation of hydrogen bonds between its phenolic hydroxyl and the amide group of these proteins.

## 4. Conclusions

In this work, the isoelectric precipitation method was used to extract protein from a unique vegetal product called date palm pollen (DPP), and a freeze-dryer was used to turn the DPP into powder. This study examined the impact of varying extraction temperatures (30 °C and 80 °C) on the surface, structural, thermal, and physicochemical characteristics of protein extracts obtained from DPP lyophilized. According to this study, extracting the protein at 80 °C increased its yield but also caused structural alterations in the DPP protein, such as the dissociation of some intermolecular bonds and a decrease in enthalpy. Although all cited properties were changed, the DPP protein at 80 °C was more surface active than that obtained at 30 °C; therefore, it can be used as a natural surfactant in food formulation. Tea polyphenol (EGCG) and DPP protein were utilized to bind in order to reduce the astringency of tea because of the good physiochemical results of DPP protein. The EGCG treatment altered or enhanced the DPP protein’s functional characteristics by revealing more of the protein’s hidden functional groups and relaxing the polypeptide chains. The particle size experiment revealed that it promoted the binding of DPP protein to EGCG at a molar ratio of 1:1. The results of the FTIR, CD, and UV absorption spectra showed that EGCG had a major impact on the secondary structure of the DPP protein that was extracted at 30 °C and 80 °C. This result lends credence to the theory that heating DPP increases the binding affinity of EGCG to DPP. This study’s conclusions are useful for lowering the astringency and bitterness of tea and enhancing the sensory experiences of tea beverages made with DPP protein at 80 °C. The results of this study were encouraging for the creation of food formulations that use DPP as an efficient carrier in the context of food items.

## Figures and Tables

**Figure 1 foods-14-00508-f001:**
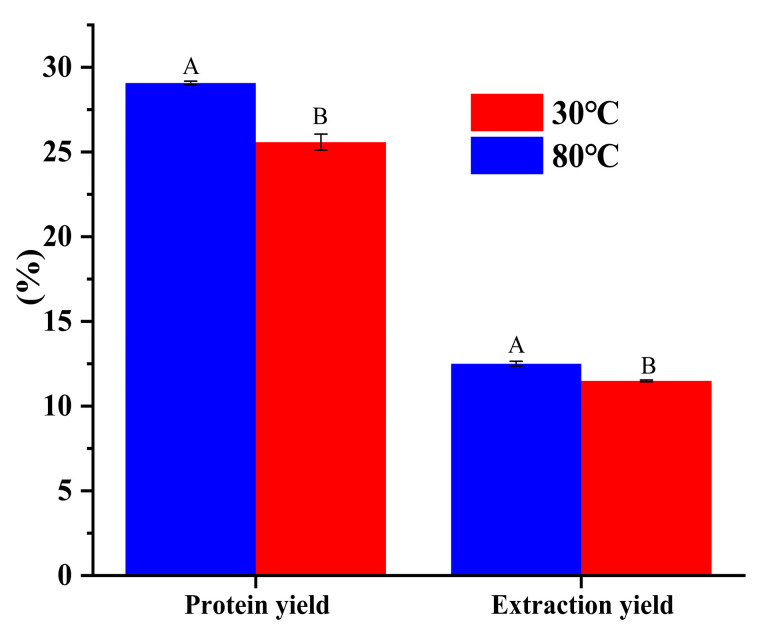
Extraction yield and protein yield of DPP at 30 °C and 80 °C. Means with the different superscript letters within the different column color are significantly different (*p* < 0.05).

**Figure 2 foods-14-00508-f002:**
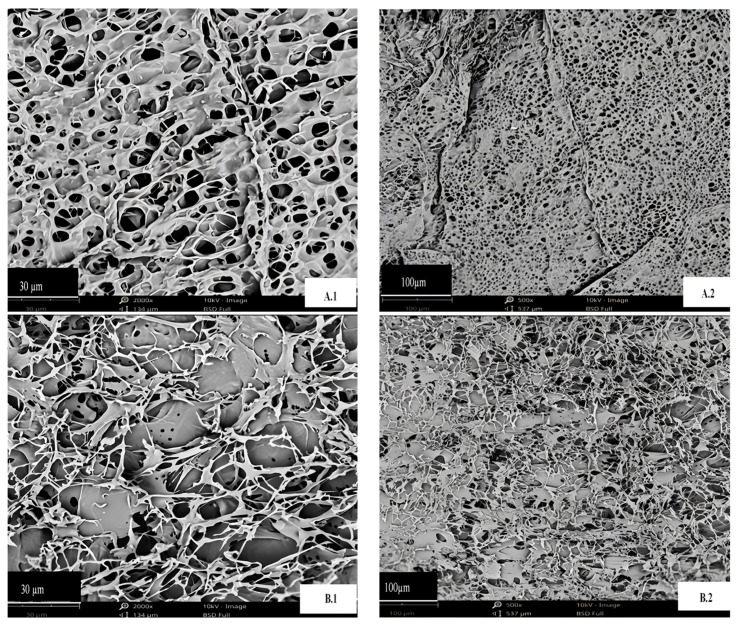
Scanning electron microscopy of DPP protein concentrates: (**A**) protein extracted with isoelectric precipitation method in 30 °C; (**B**) protein extracted with isoelectric precipitation method in 80 °C. The numbers 1 and 2 correspond to 2000- and 500-fold magnifications, respectively.

**Figure 3 foods-14-00508-f003:**
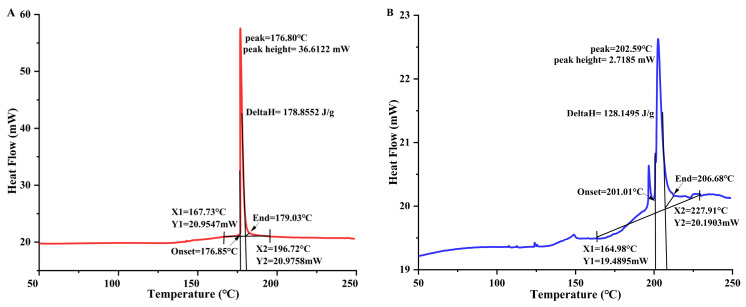
Differential scanning calorimetry of date palm pollen protein concentrates: (**A**) at 30 °C; (**B**) at 80 °C.

**Figure 4 foods-14-00508-f004:**
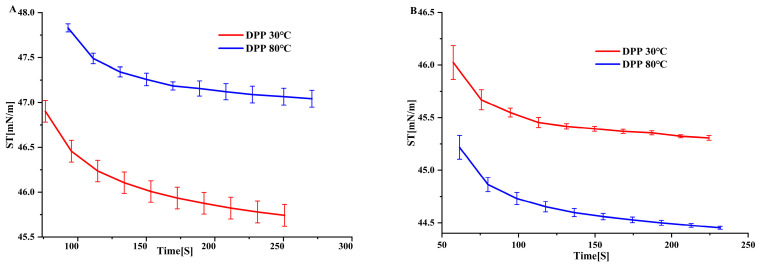
Surface tension of DPP protein concentrates obtained with isoelectric precipitation in different thermal DPP 30 °C and DPP 80 °C: (**A**) concentration—0.5 g/100 mL; (**B**) concentration—1 g/100 mL.

**Figure 5 foods-14-00508-f005:**
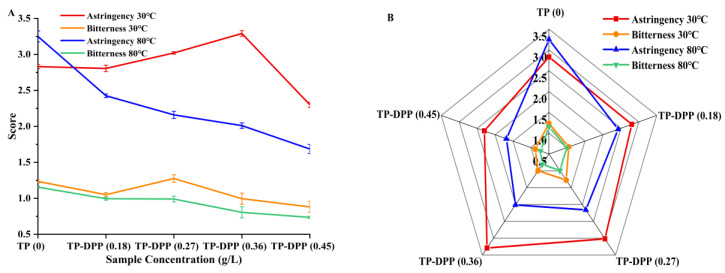
Sensory evaluation (astringency and bitterness) of TP alone and TP added to DPP protein (30 °C and 80 °C) at different concentrations (0.18, 0.27, 0.36, and 0.45 g/L). (**A**) Score plot with sensory categories; (**B**) radar plot of QDA sensory evaluation.

**Figure 6 foods-14-00508-f006:**
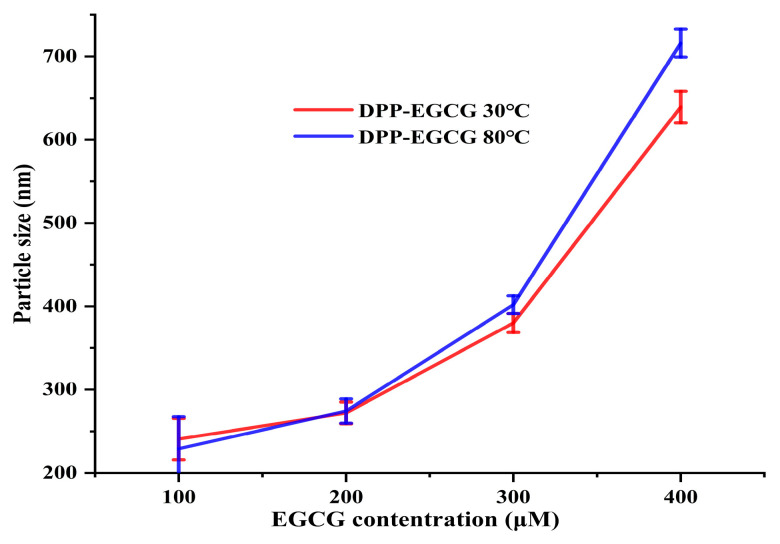
Influence of EGCG concentrations on the formation of soluble complexes with DPP at 30 °C and 80 °C.

**Figure 7 foods-14-00508-f007:**
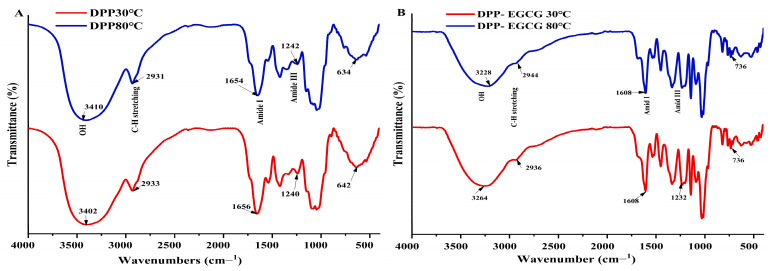
Fourier transform infrared (FTIR) spectra in the region 4000–400 cm^−1^: (**A**) DPP protein at 30 °C; 80 °C and (**B**) 1:1 ratio of freeze-dried EGCG-DPP (30 °C; 80 °C).

**Figure 8 foods-14-00508-f008:**
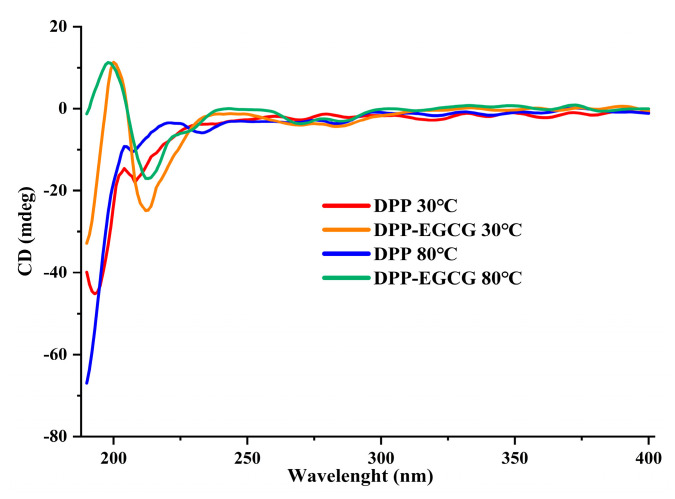
CD–visible absorption spectra of DPP protein at 30 °C and 80 °C in the absence and presence of EGCG.

**Figure 9 foods-14-00508-f009:**
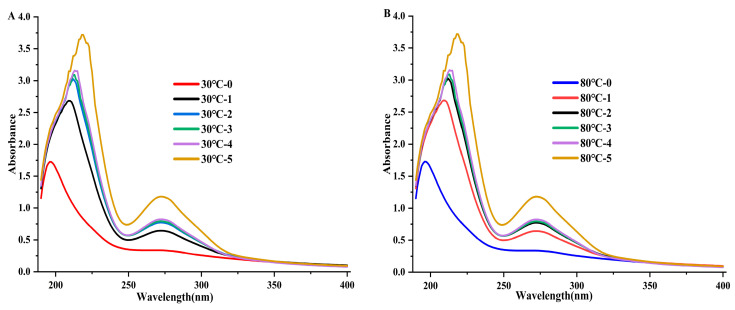
UV–visible absorption spectra of DPP at (**A**) 30 °C; (**B**) 80 °C, in the absence and presence of EGCG upon different concentrations of 0, 2.0, 4.0, 6.0, 8.0, and 10.0 × 10^−5^ mol/L.

**Table 1 foods-14-00508-t001:** Chemical properties of date palm pollen powder.

Components	Values
Moisture content (%)	7.13 ± 0.35
Ash content (%)	2.91 ± 0.13
Fiber content (%)	2.67 ± 0.22
Fat content (%)	20.00 ± 0.01
Protein content (%)	34.42 ± 0.58
Carbohydrate content (%)	32.88 ± 0.14
Polyphenols content (mg/100 g)	227.46 ± 0.16
Flavonoids content (mg/100 g)	6.37 ± 0.03

All the data are expressed as mean ± SD and are the mean values of three replicates.

## Data Availability

The original contributions presented in this study are included in the article/Appendix A. Further inquiries can be directed to the corresponding authors.

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
