# Peer review of "Effects of Extraction Temperature of Protein from Date Palm Pollen on the Astringency Taste of Tea"

_foods, 2025, doi:10.3390/foods14030508_

Round 1
Reviewer 1 Report (Previous Reviewer 2)
Comments and Suggestions for Authors
The study investigates the use of Date palm pollen protein as a plant-based alternative for reducing tea astringency, highlighting the effect of extraction temperature (30°C vs. 80°C) on protein quality and its interactions with EGCG. The manuscript is well-structured and presents comprehensive results, with conclusions well-supported by the data. Overall, the study aligns with the journal's scope and offers valuable insights. Minor suggestions for improvement are outlined below.
-The introduction should clearly emphasize the novelty of this study and include a concise review of similar works to better contextualize its unique contributions.
-The font size in Figures 3 and 5 is too small, making the text difficult to read; consider increasing it for better visibility.
-The sentence on line 225, “[10, 19, 20] These researchers presented analogous findings,” should be rephrased for clarity. Additionally, grammatical issues should be addressed throughout the manuscript, such as „Bio active“ on line 221 and „Fiber“ on line 222, „between 1620-1520 cm-1, Thus,“ on line 367, etc.
-The table in the supplementary materials is not referenced anywhere in the main text.
Author Response
For research article
Response to Reviewers
- Summary
Thank you very much for taking the time to review this manuscript. The comments and suggestions from the reviewers are very valuable. Our revised portions, according to the reviewers’ comments, are listed as follows.
This manuscript reports Effects of extraction temperature of protein from date palm pollen on the astringency taste of tea (Foods-3438494). The results are interesting, and the manuscript has been organized in a consistent manner. Hence, this work has novelty to publish in Food after major revisions by addressing the following comments:
- Point-by-point response to comments and suggestions for authors.
Reviewer 1
Comments 1: The introduction should clearly emphasize the novelty of this study and include a concise review of similar works to better contextualize its unique contributions.
Response 1: Thank you for the valuable comments. According to the comment, we revised the introduction.
This study is the first to explore how to extract protein from DPP at different temperatures. The goal is to reduce the astringency and bitterness of tea by using the interactions between protein and polyphenols. Two previous investigations examined the efficacy of DPP concentrate [16, 25]. These studies looked at how to extract protein from DPP using ultrasound at 25°C for 2 seconds, followed by separation using isoelectric precipitation at 30°C. The results showed that the protein obtained using the ultrasound method had better surface activity compared to the conventional extraction method. This protein could be used as a natural surfactant in the agri-food and pharmaceuticals fields. To our knowledge, no one has previously defined the interaction between DPP and EGCG, particularly in reducing tea astringency, and the mechanism underlying this interaction remains unknown.
Comments 2: The font size in Figures 3 and 5 is too small, making the text difficult to read; consider increasing it for better visibility.
Response 2: Thank you for pointing this out. We agree with this comment. Therefore, we revised the font size in figures.
Comments 3: The sentence on line 225, “[10, 19, 20] These researchers presented analogous findings,” should be rephrased for clarity. Additionally, grammatical issues should be addressed throughout the manuscript, such as „Bio active“ on line 221 and „Fiber“ on line 222, „between 1620-1520 cm-1, Thus,“ on line 367, etc.
Response 3: Thank you for pointing this out. We agree with this comment. Therefore, we revised the whole unclarity and grammatically incorrect sentences in the manuscript to emphasize these points.
[10, 19, 20] These researchers presented analogous findings, who found the protein content (31.11, 32.95 and 30.87) %, carbohydrate ranges from (13.41 to 35.06) % and fat content ranged from (19.31 to 20.74) % respectively.
Comments 4: The table in the supplementary materials is not referenced anywhere in the main text.
Response 4: Thanks for the expert comments, and we felt sorry for this mistake, and we mentioned the tables in a manuscript.
Reviewer 2 Report (Previous Reviewer 1)
Comments and Suggestions for Authors
I have checked the manuscript foods-3438494, which I had previously had a chance to review under the code foods-3359908. I have looked at the response given by the authors to my past suggestions, and I note that many were left unanswered/solved.
I indicate below the major aspects that need to be worked on:
. The major flaw in the manuscript remains, which is that it requires a lot of reading to understand what was done. Thus, the manuscript still requires a lot of work to be able to make out, at a first reading, that it was done.
. There is still a very bad use of units. I recommend reviewing the PDF provided in the previous review.
. It is not clear from the Abstract how EGCG and DPP were used or for what purpose. I think this section needs a lot of work, to make clear what was done, because in the current form, it does not describe the work done.
Keywords: Do not repeat words contained in the title. If you repeat words contained in the title, you lose the opportunity to increase your chances of appearing in search engines and, therefore, of being cited.
. Once an abbreviation is explained and used, it should continue to be used.
Each numerical value must be accompanied by the unit. Otherwise, it is just a number. Apply this to the whole manuscript.
L62: “L.” must not be italicized.
. Equipment and reagents are still not detailed.
L109: It remains unclear whether the same sample already discussed in the previous section was used for this test. If so, it was wrong.
. The authors' response 16 must be included in the text to understand the reason for choosing two temperatures that are so far apart.
. Clearly no significant figures are reported.
. Figure 1 is still not self-explanatory.
Author Response
For research article
Response to Reviewers
- Summary
Thank you very much for taking the time to review this manuscript. The comments and suggestions from the reviewers are very valuable. Our revised portions, according to the reviewers’ comments, are listed as follows.
This manuscript reports Effects of extraction temperature of protein from date palm pollen on the astringency taste of tea (Foods-3438494). The results are interesting, and the manuscript has been organized in a consistent manner. Hence, this work has novelty to publish in Food after major revisions by addressing the following comments:
- Point-by-point response to comments and suggestions for authors.
Reviewer 2
Comments 1: The major flaw in the manuscript remains, which is that it requires a lot of reading to understand what was done. Thus, the manuscript still requires a lot of work to be able to make out, at a first reading, that it was done.
Response 1: Thanks for your professional suggestion. We had modified our introduction (L 40-93). And we hope now it will be more understandable.
The health benefits of tea are widely recognized all over the world, but the enduring bitterness and astringency, especially in green tea, remain significant challenges to broader acceptance. These sensory qualities are primarily linked to tea's polyphenol-rich composition, encompassing flavonoids, phenolic acids, and alkaloids [1]. And catechins, particularly epigallocatechin gallate (EGCG), which is prevalent in tea leaves, significantly impact the taste profile [2].
Proteins are commonly used to decrease the bitter and astringent taste of tea. Numerous studies have examined the relationship between Tea Polyphenols (TPs) and proteins [3]. According to [4], TPs have been observed to interact with proteins that include a large concentration of alkaline amino acids, which can cause peptide tertiary structure to loosen and expose hydrophobic groups. Because of their abundance in hydroxyl groups, catechins can create hydrogen bonds with proteins' nitrogen and oxygen [5, 6]. According to most theories, there are three steps to the interaction between TPs like EGCG and proteins [7]. First, a loose, randomly coiled conformation protein binds to several sites on polyphenols, causing the protein to shrink physically and take on a tighter, spherical structure. The polyphenol complex binds to the protein surface in the second step as the concentration of polyphenols increases. Protein molecules dimerize one another, which lowers their solubility. The final stage involves the dimers' spontaneous assembly, which results in a significant quantity of particle precipitation [8].
Milk is commercially used to decrease the bitter and astringent taste of tea; however, plant-based proteins are increasingly favoured over milk-based proteins primarily because of their lower environmental impact, health benefits, and suitability for various dietary needs [9]. Moreover, plant-based proteins are typically easier to digest and are free from animal-derived hormones and antibiotics.
The pollen of the date palm (Phoenix dactylifera L.), a member of the Aceraceae family, contains more than 30% of protein [10]. And it has been widely utilized as a medicinal remedy by the ancient Egyptians and Chinese.
Processing factors like heat treatment can influence the interactions between polyphenols and proteins, as well as the structure of proteins and the biological activities of polyphenols. Many authors have noted that high heat treatments can improve the strength of polyphenol-protein interactions [11]. This raises intriguing questions about the protein part and how different extraction temperatures affect its properties. However, it is known that all proteins denature at temperatures over 40°C to 60°C. Depending on the specific proteins present and the experimental conditions, this temperature varies; higher temperatures can ensure inactive enzymes or disrupt protein-protein interactions. Therefore, in this study we employed two distinct temperatures to guarantee the interaction between protein and polyphenols.
This study is the first to explore how to extract protein from DPP at different temperatures. The goal is to reduce the astringency and bitterness of tea by using the interactions between protein and polyphenols. Two previous investigations examined the efficacy of DPP concentrate [16, 25]. These studies looked at how to extract protein from DPP using ultrasound at 25°C for 2 seconds, followed by separation using isoelectric precipitation at 30°C. The results showed that the protein obtained using the ultrasound method had better surface activity compared to the conventional extraction method. This protein could be used as a natural surfactant in the agri-food and pharmaceuticals fields. To our knowledge, no one has previously defined the interaction between DPP and EGCG, particularly in reducing tea astringency, and the mechanism underlying this interaction remains unknown.
Comments 2: There is still a very bad use of units. I recommend reviewing the PDF provided in the previous review.
Response 2: Agree. We have, accordingly, revised the units on all the manuscript to emphasize this point.
Comments 3: It is not clear from the abstract how EGCG and DPP were used or for what purpose. I think this section needs a lot of work to make clear what was done, because in the current form, it does not describe the work done.
Response 3: Thanks for the valuable suggestions. As reviewers advised, we have carefully revised the abstract of the original manuscript (pages 1-2) (L 21-36). In the revised manuscript, the clarification of how EGCG and DPP were used or for what purpose. The astringency of tea, predominantly attributed to epigallocatechin gallate (EGCG), plays a crucial role in shaping its overall quality, and plant-based proteins are gaining popularity as a preferred alternative to milk-based proteins for enhancing the flavor profile of tea. This study investigated the impact of extraction temperature on date palm pollen (DPP) protein quality and tea astringency, comparing temperatures of 30°C and 80°C. Results indicated that higher extraction temperatures yield more protein and improve the thermal and surface properties of DPP. The molecular interaction between DPP and EGCG was investigated in an aqueous solution, and spectroscopic analyses (FTIR, UV, and CD) revealed that EGCG interactions at a 1:1 molar ratio induced structural changes in α-helix and β-sheet content in secondary structures in DPP, particularly at 80°C, which strengthened and enhanced the hydrophobic interactions and hydrogen bonds between DPP molecules as EGCG concentration increased. A sensory evaluation using quantitative descriptive analysis (QDA) confirmed a significant reduction in astringency in DPP-tea polyphenol solutions extracted at 80°C. This research highlights the potential of DPP as a functional ingredient in the food industry, creating a protein-polyphenol complex that reduces tea's astringency while maintaining its unique flavor profile, thus offering a novel approach to enhance tea beverages.
Comments 4: Keywords: Do not repeat words contained in the title. If you repeat words contained in the title, you lose the opportunity to increase your chances of appearing in search engines and, therefore, of being cited.
Response 4: Thank you for the valuable comments. According to this suggestion, we changed the words in keywords (L 37-38), Keywords: Protein Structure, (-)-Eepigallocatechin gallate, Interaction, Spectrometry measurements, Sensory evaluation.
Comments 5: Once an abbreviation is explained and used, it should continue to be used.
Response 5: Agree. We have, accordingly, written the abbreviations after explaining to emphasize this point.
Comments 6: Each numerical value must be accompanied by the unit. Otherwise, it is just a number. Apply this to the whole manuscript.
Response 6: Agree. We have, accordingly, revised the units in the whole manuscript according to comment 2 to emphasize this point.
Comments 7: L62: “L.” must not be italicized.
Response 7: Thank you for pointing this out. We agree with this comment. Therefore, we changed the scientific name of the date palm tree. The pollen of the date palm tree (Phoenix dactylifera L.) (L64).
Comments 8: Equipment and reagents are still not detailed.
Response 8: Thanks for the expert comments, and we felt sorry. We added the reagents and materials in the section of materials, and the equipment’s mentioned in each method individually.
2.1. Materials
The DPP powder was procured from Xi’an Xuanqing Import & Export Co., Ltd. and maintained at 4°C until required. EGCG was provided by Shanghai Darui Fine Chemistry Co., Ltd. and subsequently stored at 4°C for future use. All remaining chemicals utilized were of analytical grade, unless specified otherwise. Sodium hydroxide and hydrochloric acid were purchased from Yuexiang Chemical Industry Company Limited (Chongqing, China). tannin and quinine monohydrochloride dehydrate were obtained from Bide Company (Shanghai, China).
2.2. proximate composition and Bio-active Compounds of date palm pollen
2.2.1. Chemical composition analysis: The study aimed to determine moisture, crude fiber, crude fat, crude protein, and ash in a sample using the official AOAC method [12]. Moisture was determined by weighing two grams of well-mixed samples in a preheated dish and heating them in an oven for 24 hours. Ash was determined by placing the sample in a crucible and heating it in a muffle furnace for 3 hours or more until white gray or reddish ash was obtained. Crude protein was determined using the micro Kjeldahl (N × 6.25) distillation method, which involved digesting the sample with a catalyst mixture and adding concentrated nitrogen-free sulphuric acid. Fat (ether extract) was determined by a Soxhlet extractor using a dry, empty extraction flask and petroleum spirit. Crude fiber was determined using a mixture of H2SO4 and pre-heated KOH, and carbohydrates were determined by difference using the equation: 100 − [moisture + crude fat + crude protein + ash + crude fiber] %.
Comments 9: L109: It remains unclear whether the same sample already discussed in the previous section was used for this test. If so, it was wrong.
Response 9: Thanks for the expert comments, and we felt sorry for this confused. Yes, the same sample extracted by a previous section was used in this section.
The Folin-Ciocâlteu method was utilized to determine the total phenolic content. A combination of 500 µL of the previously mentioned extract and 2.5 mL of a 1:10 Folin-Ciocâlteu solution in water was prepared, followed by the addition of a 2 mL solution of 7.5% sodium carbonate. Subsequently, the tubes were left at room temperature for fifteen minutes, and the absorbance was measured at 765 nm using a UV spectrophotometer (Shimadzu, China), with distilled water used as the blank sample [14]. The total polyphenol content was quantified in mg/100 g DPP and expressed as gallic acid equivalents (GAE). A standard curve using gallic acid with concentrations ranging from 0 to 50 mg/L was established [13].
Comments 10: The authors' response 16 must be included in the text to understand the reason for choosing two temperatures that are so far apart.
Response 10: Thank you for pointing this out. We agree with this comment and included it in the text. Processing factors like heat treatment can influence the interactions between polyphenols and proteins, as well as the structure of proteins and the biological activities of polyphenols. Many authors have noted that high heat treatments can improve the strength of polyphenol-protein interactions [11]. This raises intriguing questions about the protein part and how different extraction temperatures affect its properties. However, it is known that all proteins denature at temperatures over 40°C to 60°C. Depending on the specific proteins present and the experimental conditions, this temperature varies; higher temperatures can ensure inactive enzymes or disrupt protein-protein interactions. Therefore, in this study we employed two distinct temperatures to guarantee the interaction between protein and polyphenols.
Comments 11: Clearly no significant figures are reported. Figure 1 is still not self-explanatory.
Response 11: Thank you for pointing this out. We agree with this comment, and we felt sorry. Therefore, we revised the significant figures. As can be shown, for DPP 30°C and DPP 80°C, respectively, the protein yield increased from 25.583% to 29.07%, and the extraction yield increased slightly from 11.487% to 12.503%.
Round 2
Reviewer 2 Report (Previous Reviewer 1)
Comments and Suggestions for Authors
I thank the authors for their responses. I note that they have made considerable changes in the text, but I still find the text a bit difficult to understand. However, I must acknowledge that the manuscript describes a large number of results which shows a great effort on the part of the authors. This is the third revision to the paper and some of them are still not taken into account. I apologize if I have not been clear.
1. The misuse of units, e.g., hour, ml, etc., continues.
2. The phrase “previously mentioned” in L123 implies that the sample from the previous step was used.
3. Significant figures are still not reported, e.g., the values in Table 1, I consider two (2) figures to be sufficient.
4. I still don't understand Figure 1 on its own.
Author Response
For research article
Response to Reviewers
- Summary
Thank you very much for taking the time to review this manuscript for the second time. The comments and suggestions from the reviewers are very valuable. Our revised portions according to the reviewers’ comments are listed as follows.
This manuscript reports Effects of extraction temperature of protein from date palm pollen on the astringency taste of tea (Foods- 3438494). The results are interesting, and the manuscript has been organized in a consistent manner. Hence, this work has novelty to publish in Food after major revisions by addressing the following comments:
- Point-by-point response to Comments and Suggestions for Authors.
Reviewer 2
Comments 1: The misuse of units, e.g., hour, ml, etc., continues.
Response 1: Agree. We have, accordingly, revised the units on all the manuscript to emphasize this point, and we felt sorry for this misuse.
Comments 2: The phrase “previously mentioned” in L123 implies that the sample from the previous step was used.
Response 2: Thanks for the expert comments, and we felt sorry for this confused. Yes, the same sample extracted by a previous section was used in this section.
A total of 2 g of DPP were combined with 20 mL of 50% acetone and vigorously stirred for 2 h at 25 ℃ before being centrifuged for 20 min at 4500 rpm. To enhance the extraction of polyphenols and flavonoids, the process was repeated twice [13]. (this is an extraction method).
The Folin-Ciocâlteu method was utilized to determine the total phenolic content. A combination of 500 µL of the previously mentioned extract and 2.5 mL of a 1:10 Folin-Ciocâlteu solution in water was prepared, followed by the addition of a 2 mL solution of 7.5% sodium carbonate. Subsequently, the tubes were left at room temperature for 15 min, and the absorbance was measured at 765 nm using a UV spectrophotometer (Shimadzu, China), with distilled water used as the blank sample [14]. The total polyphenol content was quantified in mg/100 g DPP and expressed as gallic acid equivalents (GAE). A standard curve using gallic acid with concentrations ranging from (0 to 50) mg/L was established [13].
Comments 3: Significant figures are still not reported, e.g., the values in Table 1; I consider two (2) figures to be sufficient.
Response 3: Thank you for pointing this out. We agree with this comment, and we felt sorry. Therefore, we revised the significant figures.
Comment 4: I still don't understand Figure 1 on its own.
Response 4: Thanks for the expert comments, and we felt sorry for this confused. Fig. 1. shows that heat treatment significantly affected protein yield and extraction efficiency during DPP's isoelectric precipitation at 30°C and 80°C (P < 0.05). Protein yield increased from 25.58% at 30°C to 29.07% at 80°C, while extraction yield rose slightly from 11.49% to 12.50%. The higher temperature (80°C) likely enhanced protein release by disrupting cell wall integrity, which aligns with findings from previous studies [16, 25]. This difference is attributed to the increased release of proteins at elevated temperatures.
This manuscript is a resubmission of an earlier submission. The following is a list of the peer review reports and author responses from that submission.
Round 1
Reviewer 1 Report
Comments and Suggestions for Authors
With all due respect to the authors of the foods-3359908 work, I must point out that I found it very difficult to understand. At the moment, I do not know if it was very clear to me what was done. With the above and taking into account that this work of revision is to put me in the shoes of the future readers of the work, I invite the authors to make an effort and improve the writing of the work, so that the main objective is more understandable, and in this way to be able to appreciate the great amount of analysis that is presented.
In addition to the above, the temperature difference studied is abysmal and obvious differences are to be expected due to the mass transfer mechanisms that occur at the two very different conditions.
There are several aspects that are more urgent to clarify than others, but below, I try to give specific comments:
. There is a very bad use of units throughout the body of the paper. I attach a checklist from NIST, for use by the authors.
. Once an abbreviation is explained and used, it should continue to be used.
L18: The authors indicate “In a recent study”. This expression gives me the impression that they refer to a study other than foods-3359908.
L22: Each numerical value must be accompanied by the unit. Otherwise, it is just a number. Apply this to the whole manuscript.
. It is not clear from the Abstract how EGCG and DPP were used or for what purpose. I think this section needs a lot of work, to make clear what was done, because in the current form, it does not describe the work done.
Keywords: Do not repeat words contained in the title.
L33-36: Which theories? Not cited.
L40: describe EGCG the first time.
L70: Scientific names are written in italics.
L79: It could be used, but not as a by-product. Rather, the by-product could be put to use, or added value.
L85: Should be described in the past tense.
. No details of any equipment are given. Nor are the specifications of the reagents and materials used detailed, which greatly compromises the reproducibility of the work. In general, the M&M section should be better described in a logical order to understand the sequence of the assays and at what stage they are used. The authors could be helped by diagrams depicting the production steps. Hopefully these diagrams include specific operating variables.
. date palm pollen powder was purchased. This confused me greatly at the beginning, since, according to the title, I thought that this would be obtained by the authors. Later, and after much reading, it was that I understood that the authors actually obtain “pollen protein concentrate”. It is very difficult to understand what was done.
L111-113: all the units were misstated. This extends throughout the paper, mainly in the M&M section.
L120: 100 what of DPP? L? kg?
L124: the same extract was used for two analyses? In my experience, this should not have been the case.
L134: why these two temperatures so far apart? There is a 50°C difference. There is no reason to make such an extreme comparison.
L238: Considering what is said in L99, a supplied product is being analyzed. I do not see much sense in applying analysis to a supplied product, which most likely must have a technical data sheet from the company.
. The format of Table 1 is very bad. Additionally, significant figures should be reported.
L262: With such a difference of 50 ºC, it is almost obvious that there are statistically significant differences.
. L265: Significant figures should be reported. Extend its use to the rest of the text.
. I don't understand the difference between the two yields. If one is protein, what is the other?

Author Response
For research article
Response to Reviewers
- Summary
Thank you very much for taking the time to review this manuscript. The comments and suggestions from the reviewers are very valuable. Our revised portions according to the reviewers’ comments are listed as follows.
This manuscript reports Effects of extraction temperature of protein from date palm pollen on the astringency taste of tea (Foods-3359908). The results are interesting, and the manuscript has been organized in a consistent manner. Hence, this work has novelty to publish in Food after major revisions by addressing the following comments:
- Point-by-point response to Comments and Suggestions for Authors.
Reviewer 1
Comments 1: L18: The authors indicate “In a recent study”. This expression gives me the impression that they refer to a study other than foods-3359908.
Response 1: Thank you for pointing this out. In order to make the expression better for reading, we had changed the sentence to ''This study investigated'' (L24).
Comments 2: L22: Each numerical value must be accompanied by the unit. Otherwise, it is just a number. Apply this to the whole manuscript.
Response 2: Agree. We have, accordingly, revised the units on all the manuscript to emphasize this point.
Comments 3: It is not clear from the Abstract how EGCG and DPP were used or for what purpose. I think this section needs a lot of work, to make clear what was done, because in the current form, it does not describe the work done.
Response 3: Thanks for the valuable suggestions. As reviewers advised, we have carefully revised the abstract of the original manuscript (pages 1-2) (L 21-34). In the revised manuscript, the clarification of how EGCG and DPP were used or for what purpose. The astringency of tea, predominantly attributed to epigallocatechin gallate (EGCG), play a crucial role in shaping its overall quality, and plant-based proteins are gaining popularity as a preferred alternative to milk-based proteins for enhancing the flavor profile of tea. This study investigated the impact of extraction temperature on Date palm pollen (DPP) protein quality and tea astringency, comparing temperatures of 30°C and 80°C. Results indicated that higher extraction temperatures yield more protein and improve the thermal and surface properties of DPP. Spectroscopic analyses (FTIR, UV, and CD) revealed that EGCG interactions at a 1:1 molar ratio induced structural changes in DPP, particularly at 80°C, strengthening the bonds between DPP molecules as EGCG concentration increases. A sensory evaluation using quantitative descriptive analysis (QDA) confirmed a significant reduction in astringency in DPP-tea polyphenol solutions extracted at 80°C. This research highlights the potential of DPP as a functional ingredient in the food industry, creating a protein-polyphenol complex that reduces tea's astringency while maintaining its unique flavor profile, thus offering a novel approach to enhance tea beverages.
Comments 4: Keywords: Do not repeat words contained in the title.
Response 4: Thank you for the valuable comments. According to this suggestion, we changed the words in keywords (L 35-36), Keywords: Astringency, Date Palm Pollen Protein, (-)-Eepigallocatechin gallate, Interaction, Spectrometry measurements.
Comments 5: L33-36: Which theories? Not cited.
Response 5: Thanks for your professional suggestion. We had modified our introduction (L 38-70). The health benefits of tea are widely recognized all over the world, but the enduring bitterness and astringency, especially in green tea, remain significant challenges to broader acceptance. These sensory qualities are primarily linked to tea's polyphenol-rich composition, encompassing flavonoids, phenolic acids, and alkaloids [1]. And catechins, particularly epigallocatechin gallate (EGCG), which is prevalent in tea leaves significantly impact the taste profile [2].
Proteins are commonly used to decrease the bitter and astringency taste of tea. Numerous studies have examined the relationship between Tea Polyphenols (TPs) and proteins [3]. According to [4], TPs have been observed to interact with proteins that include a large concentration of alkaline amino acids, which can cause peptide tertiary structure to loosen and expose hydrophobic groups. Because of their abundance in hydroxyl groups, catechins can create hydrogen bonds with proteins' nitrogen and oxygen [5, 6]. According to most theories, there are three steps to the interaction between TPs like EGCG and proteins [7]. First, a loose, randomly coiled conformation protein binds to several sites on polyphenols, causing the protein to shrink physically and take on a tighter, spherical structure. The polyphenol complex binds to the protein surface in the second step as the concentration of polyphenols increases. Protein molecules dimerize one another, which lowers their solubility. The final stage involves the dimers' spontaneous assembly, which results in a significant quantity of particle precipitation [8].
Milk is commercially used to decrease the bitter and astringent taste of tea; however, plant-based proteins are increasingly favoured over milk-based proteins primarily because of their lower environmental impact, health benefits, and suitability for various dietary needs [9]. Moreover, plant-based proteins are typically easier to digest and are free from animal-derived hormones and antibiotics.
The pollen of the date palm (Phoenix dactylifera L.), a member of the Aceraceae family, contains more than 30% of protein [10]. And it has been widely utilized as a medicinal remedy by the ancient high Egyptians and Chinese.
Processing factors like heat treatment can influence the interactions between polyphenols and proteins, as well as the structure of proteins and the biological activities of polyphenols. Several authors have reported that high-temperature thermal treatments may increase the strength of the polyphenol - protein binding [11], which led to an intriguing exploration of the proteic fraction and an investigation into the influence of extraction temperature on its diverse characteristics.
Comments 6: L40: describe EGCG the first time
Response 6: Agree. We have, accordingly, write the whole abbreviation of epigallocatechin gallate (EGCG) (L42). To emphasize this point.
Comments 7: L70: Scientific names are written in italics.
Response 7: Thank you for pointing this out. We agree with this comment. Therefore, we changed the scientific name of date palm tree to italics. The pollen of the date palm tree (Phoenix dactylifera L.), a member of the Aceraceae family. (L62).
Comments 8: L79: It could be used, but not as a by-product. Rather, the by-product could be put to use, or added value.
Response 8: Agree. We have, accordingly, revised the whole introduction.
Comments 9: L85: Should be described in the past tense.
Response 10: Agree, we have changed the sentence to past tense. The objective of this study was to explore the influence of heat treatment (at 30℃ and 80 °C) on the quality of DPP protein based on its physio-chemical properties. (L71). To emphasize this point.
Comments 11: No details of any equipment are given. Nor are the specifications of the reagents and materials used detailed, which greatly compromises the reproducibility of the work. In general, the M&M section should be better described in a logical order to understand the sequence of the assays and at what stage they are used. The authors could be helped by diagrams depicting the production steps. Hopefully these diagrams include specific operating variables.
Response 11: Thanks for the expert comments, and we felt sorry for the unclear M&M. we added the reagents and materials in section of materials, and the equipment’s was already mentioned in each method individually.
- Introduction
- Materials and Methods
2.1 Materials
2.2. proximate composition and Bioactive Compounds of date palm pollen
2.2.1. Chemical composition analysis
2.2.2. Polyphenols extraction and determination
2.2.3. Determination of Flavonoids
2.3. Preparation of date palm pollen protein concentrates by isoelectric participate method
2.4. Determination of date palm pollen protein properties
2.4.1 Determination of extraction and protein yields
2.5. Scanning electronic microscopy (SEM)
2.6. Differential scanning calorimetry (DSC)
2.7. Surface tension
2.8. Sensory Evaluation
2.9. Preparation of protein solution and tea polyphenol solution
2.9.1. particle size measurements
2.10. Fourier transform infrared (FTIR) spectroscopy
2.11. Circular dichroism (CD) measurement
2.12. UV–vis absorption spectra
Comments 12: date palm pollen powder was purchased. This confused me greatly at the beginning, since, according to the title, I thought that this would be obtained by the authors. Later, and after much reading, it was that I understood that the authors actually obtain “pollen protein concentrate”. It is very difficult to understand what was done.
Response 12: Thanks for the expert comments, and we felt sorry for this confused and misunderstood. Actually, we purchased the date palm pollen as a powder, not as a protein powder, and then we extracted the protein from date palm pollen powder. We hope the information has now reached you in the right way.
Comments 13: L111-113: all the units were misstated. This extends throughout the paper, mainly in the M&M section.
Response 13: Agree. We have, accordingly, revised the units in whole the manuscript according to the comment in L 22 to emphasize this point.
Comments 14: L120: 100 what of DPP? L? kg?
Response 14: Thank you for pointing this out. we used 1:10 (w:v) DPP to water means that DPP by kg and water by L.
Comments 15: L124: the same extract was used for two analyses? In my experience, this should not have been the case.
Response 15: Thank you for the valuable comments. I am sorry, I couldn’t understand the question enough, but from our understanding, we used two extracts, one at 30℃ and the another at 80℃, to determine the extraction and protein yield.
Comments 16: L134: why these two temperatures so far apart? There is a 50°C difference. There is no reason to make such an extreme comparison.
Response 16: Thank you for pointing this out. We agree with this comment. It's known for all the temperature of protein denaturation, typically above 40℃ to 60℃, and this depends on specific proteins present and the condition of the experiment, and to ensure inactive enzymes or disrupt protein-protein interaction, higher temperatures may be used. Processing factors like heat treatment can influence the interactions between polyphenols and proteins, as well as the structure of proteins and the biological activities of polyphenols. Several authors have reported that high-temperature thermal treatments may increase the strength of the polyphenol-protein binding. Therefore, we used this temperature difference of 50℃ to ensure the interaction between protein and polyphenols.
Comments 17: L238: Considering what is said in L99, a supplied product is being analyzed. I do not see much sense in applying analysis to a supplied product, which most likely must have a technical data sheet from the company.
Response 17: Thanks for your valuable comment. According to L99, the date palm pollen we purchased without any technical data sheet, means that the company did not provide us the data sheet of chemical composition of DPP, and also, we did the analysis to ensure protein content and other contents.
Comments 18: The format of Table 1 is very bad. Additionally, significant figures should be reported.
Response 18: Thank you for pointing this out. We agree with this comment. Therefore, we changed the Table 1 format.
Table 1: Chemical Properties of Date Palm Pollen Powder
|
Components |
Values |
|
Moisture content (%) Ash content (%) Fiber content (%) Oil content (%) Protein content (%) Carbohydrate content (%) Polyphenols content (mg/100g) Flavonoids content (mg/100g) |
7.13 ±0.35 2.91 ±0.13 2.67 ±0.22 20.00 ±0.01 34.42 ±0.58 32.88 ±0.14 227.46 ±0.16 6.37 ±0.03 |
All the data are expressed as mean ±SD and are the mean of three replicates.
Comments 19: L262: With such a difference of 50 ºC, it is almost obvious that there are statistically significant differences.
Response 19: Agree, we find a vary statistically significant difference (P < 0.05) between the temperatures, specifically in protein yield. (L 256) Fig. 1 emphasizes this result.
Comments 20: L265: Significant figures should be reported. Extend its use to the rest of the text.
Response 20: Thank you for pointing this out. We agree with this comment. Therefore, we revised the significant figures.
Comments 21: I don't understand the difference between the two yields. If one is protein, what is the other?
Response 21: Thanks for the expert comments, and we felt sorry for this confused and misunderstood, the two samples of protein yield, one is protein yield at 30℃ and other at 80℃.
Reviewer 2 Report
Comments and Suggestions for Authors
The manuscript is a valuable contribution to the field of functional beverages, providing insights into reducing tea astringency using plant-based proteins. The study is well-structured and clearly written, presenting substantial experimental data. However, the following suggestions could further improve the quality of the manuscript:
- In the abstract, replace "In a recent study" with "In this study"
-Include a brief overview of previously studied methods for extracting DPP protein, highlighting the temperatures used and explaining the rationale for selecting these specific methods.
-Clarify why the temperatures of 30°C and 80°C were specifically chosen for the experiments.
-Ensure that Table 1 is properly aligned to enhance readability and presentation.
-Pay attention to figure captions: In Figure 1, it is unclear what the labels "A" and "B" represent on the graph. In Figure 2, A1 and B1 correspond to 2000-fold magnification, while A2 and B2 correspond to 500-fold magnification. However, the caption incorrectly states the reverse (“1,2 correspond to 500 and 2000-fold magnifications”). In Figures 5 and 9, the labels "A" and "B" are used, but their meanings are not specified in the captions.
-Improve the resolution and overall quality of the figures
Author Response
For research article
Response to Reviewers
- Summary
Thank you very much for taking the time to review this manuscript. The comments and suggestions from the reviewers are very valuable. Our revised portions according to the reviewers’ comments are listed as follows.
This manuscript reports Effects of extraction temperature of protein from date palm pollen on the astringency taste of tea (Foods-3359908). The results are interesting, and the manuscript has been organized in a consistent manner. Hence, this work has novelty to publish in Food after major revisions by addressing the following comments:
- Point-by-point response to Comments and Suggestions for Authors.
Reviewers 2
Comments 1: In the abstract, replace "In a recent study" with "In this study"
Response 1: Thanks for your suggestion. We agree with this comment. Therefore, we had changed the sentence to '' this study investigated''.
Comments 2: Include a brief overview of previously studied methods for extracting DPP protein, highlighting the temperatures used and explaining the rationale for selecting these specific methods.
Response 2: Recently, two studies looked at how well date palm pollen concentrate works [Sebii, et al. 2019, Karra et al. 2020]. These studies focused on getting protein extract from date palm pollen by using ultrasound treatment at 25℃ for 2 seconds and isoelectric precipitation at 30℃.
Comments 3: Clarify why the temperatures of 30°C and 80°C were specifically chosen for the experiments.
Response 3: Thank you for pointing this out. We agree with this comment. It's known for all the temperature of protein denaturation, typically above 40℃ to 60℃, and this depends on specific proteins present and the condition of the experiment, and to ensure inactive enzymes or disrupt protein-protein interaction, higher temperatures may be used. Processing factors like heat treatment can influence the interactions between polyphenols and proteins, as well as the structure of proteins and the biological activities of polyphenols. Several authors have reported that high-temperature thermal treatments may increase the strength of the polyphenol-protein binding. Therefore, we used this temperature difference of 50℃ to ensure the interaction between protein and polyphenols.
Comments 4: Ensure that Table 1 is properly aligned to enhance readability and presentation.
Response 4: Thank you for pointing this out. We agree with this comment. Therefore, we changed the Table 1 format.
Table 1: Chemical Properties of Date Palm Pollen Powder
|
Components |
Values |
|
Moisture content (%) Ash content (%) Fiber content (%) Oil content (%) Protein content (%) Carbohydrate content (%) Polyphenols content (mg/100g) Flavonoids content (mg/100g) |
7.13 ±0.35 2.91 ±0.13 2.67 ±0.22 20.00 ±0.01 34.42 ±0.58 32.88 ±0.14 227.46 ±0.16 6.37 ±0.03 |
All the data are expressed as mean ±SD and are the mean of three replicates.
Comments 5: Pay attention to figure captions: In Figure 1, it is unclear what the labels "A" and "B" represent on the graph. In Figure 2, A1 and B1 correspond to 2000-fold magnification, while A2 and B2 correspond to 500-fold magnification. However, the caption incorrectly states the reverse (“1,2 correspond to 500 and 2000-fold magnifications”). In Figures 5 and 9, the labels "A" and "B" are used, but their meanings are not specified in the captions.
Response 5: Thank you for the valuable comments. According to the comment, we revised the figures label. Figure 1, A & B Means with the different superscript letters within the same column color are significantly different (P < 0.05). Figure 2, we modified to the number 1, 2 corresponds to 2000 and 500-fold magnifications, respectively. Figure 5, (A) score plot with sensory categories; (B) radar plot of QDA sensory evaluation. Figure 9, UV–visible absorption spectra of DPP at (A) 30℃; (B) 80℃, in the absence and presence of EGCG upon different concentrations of 0, 2.0, 4.0, 6.0, 8.0, and 10.0 × 10−5 mol/L.
Comments 6: Improve the resolution and overall quality of the figures
Response 6: Agree. We have, accordingly, revised the resolution and overall quality of the figures